# Tissue Expansion Improves the Outcome and Predictability for Alveolar Bone Augmentation: Prospective, Multicenter, Randomized Controlled Trial

**DOI:** 10.3390/jcm9041143

**Published:** 2020-04-16

**Authors:** Soo-Hwan Byun, Sun-Hyun Kim, Sura Cho, Ho Lee, Ho-Kyung Lim, Ju-Won Kim, Ui-Lyong Lee, Wan Song, Sun-Jong Kim, Min-Kyoung Kim, Jin-Woo Kim

**Affiliations:** 1Department of Oral and Maxillofacial Surgery, Sacred Heart Hospital, Hallym University Medical Center, Kyonggi-do 14068, Korea; purheit@daum.net (S.-H.B.); kjw9199@gmail.com (J.-W.K.); 2Research Society of Gangnam Oral and Maxillofacial Surgeons, Seoul 07985, Korea; nbeauty@naver.com (H.L.); ungassi@gmail.com (H.-K.L.); davidjoy76@gmail.com (U.-L.L.); 3Department of Oral and Maxillofacial Surgery, College of Medicine, Ewha Womans University, Seoul 07985, Korea; lovegusdl@gmail.com (S.-H.K.); sura.cho0104@gmail.com (S.C.); sjsj7777@ewha.ac.kr (S.-J.K.); 4Department of Oral and Maxillofacial Surgery, Section of Dentistry, Seoul Metropolitan Government – Seoul National University Boramae Medical Center, Seoul 07061, Korea; 5Department of Oral and Maxillofacial Surgery, Korea University Medical Center, Guro Hospital, Seoul 08308, Korea; 6Department of Oral and Maxillofacial Surgery, Chung-Ang University Hospital, Seoul 06973, Korea; 7Department of Urology, Samsung Medical Center, Sungkyunkwan University School of Medicine, Seoul 06351, Korea; waniyo25@hanmail.net; 8Bio R&D Center, Osstem implant Co., Ltd. Busan 07789, Korea; min1107@osstem.com

**Keywords:** tissue expansion, bone regeneration, clinical trial, graft, dental implant

## Abstract

Objectives: The purpose of this study was to evaluate the effectiveness of the intraoral use of subperiosteally placed self-inflating tissue expanders for subsequent bone augmentation and implant integrity. Material and methods: A prospective, multicenter, randomized controlled trial was performed on patients requiring alveolar bone graft for dental implant insertion. Patients were assigned to three groups: tissue expansion and tunneling graft (TET group), tissue expansion and conventional bone graft (TEG), and control group without tissue expansion. Dimensional changes of soft tissue and radiographic vertical bone gain, retention, and peri-implant marginal bone changes were evaluated and secondary outcomes; clinical complications and thickness changes of expanded overlying tissue were assessed. Results: Among 75 patients screened, a total of 57 patients were included in the final analysis. Most patients showed uneventful soft tissue expansion without any inflammatory sign or symptoms. Ultrasonographic measurements of overlying gingiva revealed no thinning after tissue expansion (*p* > 0.05). Mean soft vertical and horizontal tissue measurements at the end of its expansion were 5.62 and 6.03 mm, respectively. Significantly higher vertical bone gain was shown in the TEG (5.71 ± 1.99 mm) compared with that in the control patients (4.32 ± 0.97 mm; *p* < 0.05). Hard tissue retention— measured by bone resorption after 6 months—showed that control group showed higher amount of vertical (2.06 ± 1.00 mm) and horizontal bone resorption (1.69 ± 0.81 mm) compared to that of the TEG group (*p* < 0.05). Conclusion: The self-inflating tissue expander effectively augmented soft tissue volume and both conventional bone graft and tunneling techniques confirmed their effectiveness in bone augmentation. With greater amount of bone gain and better 6 month hard tissue integrity, the TEG group compared to the control group—without tissue expansion—showed that the combined modality of tissue expander use and guided bone regeneration (GBR) technique may improve the outcome and predictability of hard tissue augmentation.

## 1. Introduction

Soft tissue deficiency may occur due to various causes including tumor resection, trauma, congenital problems, etc. Restoration of soft tissue deficiency in such circumstances is a challenge to clinicians, but they may be resolved with local flaps, pedicled flaps, free flaps, and other substitutes such as allograft and alloplastic graft [1]. An alternative method to flap exposure or graft is a tissue expansion. The concept of tissue expansion was first postulated by Neumann as a method of expanding local skin for ear reconstruction in 1957 [2]. After about 20 years, the concept of tissue expansion was brought into attention by the study of Radovan et al. [3], who later with Austad and Rose, developed a self-inflating tissue expander [4]. A tissue expander reduces the need for free tissue graft, minimizing the morbidity of donor site in addition to preserving the texture, color, and sensation of local soft tissue [5].

Most tissue expanders are designed and made using inflatable balloons or osmotically active hydrogel, and the rate of expansion is controlled by fluid entry or various porosities [6,7,8]. The principal concept of the tissue expansion technique is its progressive enlargement. The volume of the implanted tissue expander should increase gradually, maintaining constant mechanical strength against the surrounding tissue for a certain period of time [9]. The tissue expander is usually placed below the periosteum and the skin and the enlarged tissue expander exerts pressure, which induces the growth of the soft tissue. This process is continued until maximum expansion of the implanted expander has been reached, and the soft tissue generally thins, and re-alignment of collagen fibers occur.

Contraindications to tissue expansion include unwillingness or medical inability to undergo multiple operations or to comply with the numerous visits required for the expansion process, noncompliance, mental disability, poorly vascularized tissues from radiation therapy, infection or open wounds, and ongoing chemotherapy [10]. Tissue expansion in infants and children is not recommended and their possible complications include thinning and deformation of the underlying bone and subsequent scar widening with growth [11]. It is reported that about 8.73% of complications arising from tissue expansion occur in the head and neck area; but, given its sufficient blood supply and resistance to infection, it is thought to be a suitable location for tissue expansion [12].

Severe atrophy of the alveolar bone is common in patients with periodontitis. A vertical alveolar bone defect can be recovered with ridge augmentation but often results in clinical problems such as wound dehiscence and infection. Deficiency of soft tissue must be solved with an alternative soft tissue management [13]. Clinicians frequently face challenges with “split-thickness” flap techniques and releasing incisions for soft tissue deficiencies [14]. This technique often causes surrounding tension from sutures on a thin mucosal flap, inducing soft tissue necrosis and wound dehiscence. It is estimated that up to 45% of attempted ridge augmentations are lost or significantly compromised when using “split-thickness” technique for bone grafts compared to a 4% bone graft exposure when using a tissue expander [15,16,17]. This warrants the use of tissue expanders in the dental field [18].

The use of a hydrogel-type, self-inflating soft tissue expander prior to bone graft can improve soft tissue management and facilitate successful settlement of the graft. In our previous pilot study, we were able to find that enough soft tissue volume was gained with the use of a self-inflating tissue expander [19]. This combined treatment modality—alveolar soft tissue expansion and subsequent bone graft—showed favorable augmented tissue integrity with minimal side effects. Therefore, the purpose of this study was to evaluate the effectiveness of the intraoral use of subperiosteally placed self-inflating tissue expanders for subsequent bone augmentation and implant integrity.

## 2. Material and Methods

### 2.1. Study Design

This study was designed to test the effectiveness of a tissue expander in soft tissue augmentation and its sequential hard tissue augmentation. The study was conducted as a multicenter randomized controlled trial (RCT) between six institutions: Ewha Woman’s University Medical Center (IRB; Institutional review board no. EUMC 2016-11-003-1), Hallym University Dongtan Sacred Hospital (IRB no. HDT 2017-02-190-001), Seoul National University Borame Medical Center (IRB no. 20170321/16-2017-18/042), Korea University Medical Center (IRB no. 2018GR0045), Chung-Ang University Hospital (IRB no. 1781-005-292), and Hallym University Sacred Heart Hospital (IRB no. 2018-I002). The study was conducted in accordance with the Declarations of Helsinki, internationally accepted guidelines for RCTs, and the CONSORT (consolidated standards of reporting trials) statement. This research was conducted by the same team of researchers as an extension of the pilot study on the clinical feasibility of the novel tissue expander [19]. After confirming its feasibility, the authors additionally included a group using the tissue expander and the guided bone regeneration (GBR) method (TEG) apart from the original TET and control (conventional GBR) groups to allow comparison between the study groups and its true control. In addition, histologic assessment and microCT analysis were systematically conducted.

Study participants were selected as patients who visited one of the six participating hospitals between November 2016 and December 2019 with severe vertical alveolar deficiency and required vertical bone augmentation for subsequent implant placement. All study participants were informed of the purpose of study and experimental protocol and written and oral consent was obtained from all enrolled patients. The inclusion criteria were adults 20–75 years of age (inclusive) with complete jawbone growth and patients with one or more missing teeth with severe vertical alveolar atrophy. The exclusion criteria were pregnancy, presence of metabolic diseases such as uncontrolled diabetes, intake of medication that may affect bone metabolism (bisphosphonate, rhPTH, denosumab, etc.), presence of uncontrolled periodontitis, uncontrolled gingivitis, or uncontrolled dental caries, history of radiation therapy in the head and neck area, presence of hemorrhagic diseases or intake of anticoagulant, allergy to implant material, smoking, and all other patients considered unsuitable for the study by the clinician for various reasons. Randomization was performed on SAS 9.2 (SAS Institute Inc., Cary, NC, USA) to divide total subjects into three trial groups with an allocation ratio of 1:1:1. Clinical trial procedures at each visit are summarized in Figure 1.

For soft tissue expansion, a hydrogel-type, self-inflating soft tissue expander (TissueMax^®^, Osstem, Seoul, Korea) was used. The tissue expander is composed of methyl methacrylate and 1-vinyl-2-pyrrolidone enclosed by a silicone envelope. Four types of expanders varying in volume and design were used accordingly: TEX007 (initial volume 0.15 mL, expanded volume 0.70 mL), TEX010 (initial volume 0.2 mL, expanded volume 1.0 mL), TEX007S2 (initial volume 0.15 mL, expanded volume 0.7 mL), and TEX010S2 (initial volume 0.2 mL, expanded volume 1.0 mL). The final expanded volume is the total expanded volume after 28 days. Due to dehiscence, all study subjects changed to using an improved version of the expander early in the study. The improved expander reduced the initial expansion rate to 64% of the original, greatly improving the healing period of the incision during the initial expansion.

### 2.2. Surgical Procedure

Patients were assigned to one of three study groups: TET (tissue expander–tunneling), TEG (tissue expander–guided bone regeneration; GBR), and control (no expander–GBR) (Figure 2).

All surgical procedures were performed under local anesthesia. In the TET and TEG groups—the tissue expansion group—a vertical incision of 5–10mm from crest to buccal side was made at two—mesial and distal—points with respect to the defective area (visit II). The mucoperiosteum flap was carefully reflected from the bone to form a “tunnel” between the two incisions [20]. A self-inflating soft tissue expander was placed though the created space and fixated with screws to prevent movement or dislocation. The incisions were sutured with 4-0/5-0 Dermalon^®^ monofilament nylon. Suture was removed 14 days post tissue expander insertion (visit III). After 4 weeks post tissue expander insertion, bone augmentation was performed simultaneous to expander removal (visit IV).

In the TET group, incision was made in the same area as that made for expander fixation. After identifying the expander and its fixing screws, they were removed. Subsequently, the collagen membrane tailored to the size of the removed expander (Cytoplast^®^ RTM collagen membrane, Osteogenics, NJ, USA; OssMem^®^, Osstem, Seoul, Korea) was carefully placed below the periosteum through the space in which the expander was removed and bone was grafted under the membrane using xeno-bone graft material (Bio-Oss^®^, Geistlich Pharma, Switzerland). Both sides of the incision were sutured and removed 14 days later (visit V).

In the TEG group, unlike TET group in which tunneling graft was performed, bone graft was done in the conventional vertical GBR method. The defect size and type were evaluated, then a d-PTFE titanium-reinforced membrane (Cytoplast^®^ Ti-250XL; Osteogenics, NJ, USA) was trimmed and applied to the defective area. The membrane was placed from the lingual side prior to bone grafting, then was folded over from the lingual to the buccal side after bone graft and fixated using metal pins when necessary. In case of insufficient graft stability, additional tenting screws were applied. After a periosteal releasing incision was made to ensure primary closure of the flap, the area was re-sutured with 4-0/5-0 monofilament nylon and removed 14 days later.

The control group followed the same procedure as the TEG group but without tissue expander insertion. A periosteal releasing incision was made to ensure primary closure of the flap in all cases of the control group. Following bone graft, a d-PTFE titanium-reinforced membrane (Cytoplast^®^ Ti-250XL; Osteogenics, NJ, USA) was placed over the grafted area before flap closure. After each surgical procedure, all patients were instructed of the post-operation guidelines and were prescribed daily chlorhexidine mouthwash (0.2%). Any wound dehiscence, perforations, inflammation, infection, or other postoperative complications were documented at every visit.

At the 6 month follow-up from the day of bone graft, patients of all groups received implant placement. Straumann (Straumann ^®^ bone level SLActive^®^, Basel, Switzerland) and Osstem (Osstem, Seoul, Korea) implants were used. While drilling for implant placement, a specially designed 1.8 mm inner diameter trephine bur was used to collect the internal bone tissue for histological analysis.

### 2.3. Data Collection and Outcome Measures

#### 2.3.1. Soft Tissue Expansion

Outcome measures were collected at visit I (pre-operative), visit IV (tissue expander removal), and at visit VI (6 months after bone graft). Cast models were scanned with a 3D scanner (Trios3^®^, 3Shape, Denmark) and transferred into a database integrated software (Geomagic Control X, 3D systems, USA) to measure soft tissue dimensional changes (Figure 3A). Scanned 3D images were exported as STL files (standard tessellation language), and each patient’s images at different time points were superimposed using adjacent teeth from the defective site as the reference point and landmark as the best-fit initial alignment [21,22]. Three vertical cross sections and three horizontal cross sections were measured to observe the horizontal and vertical changes. For the vertical change, a cross-section was drawn as a plane crossing the central fossa, then two parallel planes 3 mm right and left of the central plane were drawn at the buccal and lingual sides, respectively. Horizontal dimensional change was measured with points 2, 4, and 6 mm from the cemento-enamel junction of adjacent teeth. The RMS (root mean square) estimate values were recorded for each pair, and their mean changes were calculated.

#### 2.3.2. Bone Volume and Retention

Bone volume was recorded using CBCT (cone beam computed tomography) scans taken at visit I (pre-operative), visit IV (immediately following bone graft), visit VI (6 months after bone graft), and at visit VII (follow-up). Raw data (DiCOM format) were imported to a 3D software program (OnDemand 3D^®^, Cybermed, Seoul, Korea) and the horizontal and vertical hard tissue dimensional changes were measured through image automatic superimposition. Height and width were measured as primary outcomes of the same cross-sectional plane and the average change of three points at 2 mm intervals from the highest point down was calculated. The amount of hard tissue augmentation was calculated by subtracting the measurements of visit I from those of visit IV and the amount of bone retention was calculated by subtracting the measurements of visit IV from those of visit VI. To measure the peri-implant marginal bone resorption after prosthetic loading, the distance from the most coronal margin of the implant collar to the most coronal point of bone-to-implant contact was measured from the paraxial view of each implant at visit VII. Measurements of the mesial and distal bone crest level adjacent to each implant were made to the nearest 0.01 mm and averaged at patient level.

#### 2.3.3. Thickness of Expanded Gingiva

Ultrasound images were obtained using E-CUBE 9 Diamond imaging system (Alpinion medical systems^®^, Seoul, Korea) at visit IV. An intraoral transducer IO3-12 (3~12 MHz frequency) was used to collect the thickness of the attached gingiva as well as possible thinning of the mucosa (Figure 3B). 

#### 2.3.4. Histologic Evaluation

Bone was harvested using a 1.8 mm diameter trephine bur during the implant procedure 6 months post bone graft for histological evaluation. Bone samples were fixed in 10% phosphoric acid-buffered formalin for one week and demineralized at 4 °C with 10% EDTA (pH 7.2) before paraffin immersion. To observe the presence of regenerated bone on the basis of: (1) presence of regenerated bone and (2) residual bone graft, deparaffinization, sectioning, and staining with HE (haematoxylin and eosin) and MT (Masson’s trichrome) were performed (Figure 4). Image capture was performed using Eclipse 50i optical microscope (Nikon, Tokyo, Japan) equipped with CCD camera (MicroPublisher 3.3 RTV cooling, QImaging, Bethesda, MD, USA) and Image Pro Capture Kit platform (Media Cybernetics, Bethesda, MD, USA). Tissue trait measurements were evaluated using Image Pro Plus 6.2 software (Media Cybernetics, Bethesda, MD, USA).

#### 2.3.5. Statistical Analysis

The null hypothesis to be tested was that the mean difference in vertical bone augmentation would be the same between all groups. Using G*power 3.1 software, with test family-F-test, test statistic-ANOVA: fixed effects, omnibus, one-way, and input parameters of effect size 0.5, α error 0.05, power 0.95, and 3 groups, a sample size of 66 was obtained with 15% drop rate considered. The assumptions of normality and homogeneity of variance were confirmed, and thus further statistical testing was performed using one-way ANOVA test and independent t-test. Two researchers (S.H. Byun and S.H. Kim) independently carried out measurements and the inter-examiner agreement was calculated as the intraclass correlation coefficient (ICC) from a two-way random model and absolute agreement type. The inter-examiner ICC was 0.81 (total amount of bone gain of hard tissue measurements, 95% confidence intervals: 0.69–0.91) showing favorable agreement. A *p*-value < 0.05 was considered statistically significant and all statistical analyses were performed using software SPSS (IBM Corp., Armonk, NY, USA).

## 3. Results

A total of 75 patients were screened for the study, and 66 patients were considered suitable from which 9 dropped out. A total of 57 patients were included in the final analysis. Further details on descriptive statistics can be found in Table 1. TET and TEG groups received an insertion of self-inflating osmotic expanders (mean 31.4 days) before bone augmentation. Mean soft tissue measurements at the end of its expansion (visit IV) were similar in both TET and TEG groups with its average vertical and horizontal expansion being 5.62 and 6.03 mm, respectively. Ultrasonographic measurements of gingival thickness overlying the expanders revealed that mucosa was not thinned post tissue expansion (pre-expansion thickness 1.24 ± 0.24 mm; post-expansion thickness 1.08 ± 0.34 mm, *n* = 27, *p* > 0.05), indicating minimal clinical risk of dehiscence during the expansion.

The average amount of bone augmentation showed significant group difference, especially in vertical measurements (*p <* 0.05). Vertical bone gain in TEG (5.71 ± 1.99 mm) was significantly higher than that of the control (4.32 ± 0.97 mm). Horizontal amount of augmentation was presented with no group difference. Hard tissue retention—measured by bone resorption after 6 months (visit VI)—showed significant group difference in both vertical and horizontal measurements. Control group showed higher amount of vertical (2.06 ± 1.00 mm) and horizontal bone resorption (1.69 ± 0.81 mm) compared to that of the TEG group (*p <* 0.05; 1.16 ± 0.62 mm and 0.94 ± 0.57 mm). TET group also showed less bone resorption; however, the results were not statistically significant (*p* > 0.05). Further details can be found in Table 2. At the follow-up assessment (visit VII), the marginal bone resorption around implants was 0.43 ± 0.30 mm, 0.36 ± 0.32 mm, and 0.35 ± 0.18 mm each for TEG, TET, and control groups, respectively, and those were not statistically different.

Most patients showed uneventful soft tissue expansion without any inflammatory sign or symptoms. One patient from the TET group presented with severe scarring due to several grafting failures and showed mucosal perforation during the expansion at day 18. Tissue expander was immediately removed, and bone graft was performed. This patient later showed unreasonably high bone resorption (visit VI–IV). Two patients from the TET group and one from the TEG group showed tissue over-expansion owing to silicon envelope tearing, but they healed without dehiscence.

The histological sections taken after 6 months of GBR for each group are shown in Figure 4. All groups presented newly formed and substituted bone, residual graft materials, and non-inflammatory vascularized connective tissues. The grafts were surrounded by lamellar bone showing favorable osteogenesis and osteoconduction. Histomorphometric analyses including B.Ar/T.Ar, N.Ob/B.S, and N.Oc/B.S revealed no significant differences between the groups. Moreover, microCT evaluations revealed no significant differences between micro-architectural parameters, indicative of new bone regeneration BV/TV, BS/BV, Po(tot), and BS density (BS/TV), of the two groups (*p* > 0.05).

## 4. Discussion

Vertical ridge augmentation and soft tissue management can be a challenge to clinicians, but if necessary, expansion of oral mucosa and gingiva should be considered. There are three types of tissue expanders: standard round, rectangular, and crescent, of which the rectangular shape is the most effective. Rectangular expanders gain about 38% tissue area of the calculated surface increase of the expander, whereas round type expanders gain 25% of the calculated surface increase [23]. We used the rectangular self-inflating tissue expander which was composed of methyl methacrylate and 1-vinyl-2-pyrrolidone enclosed with a silicone envelope. The tissue expander was a hydrogel block without membrane coverage. This type of tissue expander is useful in oral and maxillofacial area in terms of design and shape.

Van Rappard et al. presented that a large surface area could increase the number of expanders with different size and shape [24]. The study recommended that when using a rectangular or crescentic expander, a proper expander would have 2.5 times larger base surface than surface of the deficiency [24]. Moreover, another study presented that the expander must have sufficient volume which is two to three times the width of the deficiency since the apical circumference of the dome of soft tissue should overlie the fully inflated expander [25].

Tissue expanders based on gradually swelling hydrogels that are permeable to hydrophilic compounds also allow the incorporation and subsequent release of pharmacologic agent such as anesthetics or antibiotics [26]. This aspect is conveniently utilized to reduce the patient’s pain and discomfort during the expansion period with medication use. von See et al. reported that tissue expansion increases microvessel density above the grafted material and osseointegration 19 days after expander placement [27]. This study showed that the use of tissue expanders may reduce morbidity and side effects. Our clinical study showed that tissue expander successfully increased vertical bone dimensions in TEG group. In a single case from the TET group, wound dehiscence and grafting failure occurred during the expansion period. The tissue expander was removed early, and bone graft was performed. The patient had severe scarring in the preoperative examination which would make high resistance against the inflation of the surrounding flap.

Various tissue responses, especially for the underlying bone have been described in many studies. Some studies reported bone resorption and reduction of bone density, while others reported no bone resorption [28,29]. Sato et al. concluded that osteoclastic bone resorption was influenced by applied pressure [30]. There was no bone resorption with a continuous compressive pressure of <1.96 kPa, however, a continuous compressive pressure of >6.86 kPa resulted in significant resorption [30]. We believe the pressure of expansion may play a role in bone resorption below the expander [31]. Due to such reasons, the tissue expanding GBR group (TEG) presented lower bone resorption and hard tissue retention, as shown with significant group differences in vertical and horizontal measurements 6 months post treatment.

On the other hand, the amount of bone resorption measured 6 months after surgery was not significant in the TET and control groups. It is predicted that this is because in TEG group, ideal ossification of graft materials can be predicted by the use of titanium reinforcement, but in comparison, the TET group has lower graft stability and a relative decrease in ossification is expected. In fact, in some TET group cases, a very ideal augmentation result was achieved, while in some cases, severe resorption of augmented bone was observed. Many attempts are necessary to improve the reliability of the TET technique. Considering the multiple drawbacks of the conventional GBR techniques compared to the TET method, such as high failure rates, risk of dehiscence, infection and resorption, relatively high cost, long procedure time, etc., the improvement of technical reliability and establishment of an appropriate indication for application protocol can make TET a very efficient and effective method of treatment.

Regarding the immune reaction, a fibrous capsule has been known to surround the expander as one of the late foreign body reactions. Hydrophobic materials induce stronger foreign body reactions than hydrophilic materials [32]. The capsule reaches its maximum thickness after about 2–2.5 months after expander insertion [33]. The capsule is composed of a double layer consisting of an inner layer of macrophages and an outer layer of fibroblasts and some lymphocytes within 7 days [33]. The outer layer changes to rich collagen fibers, and the inner layer adjacent to the capsule richly vascularizes [33]. Due to this foreign body reaction against the expander, the expanded soft tissue shows appropriate texture and thickness in this study. Even though the soft tissue was fully expanded, the surrounding soft tissue did not lose its thickness. This capsule can help reduce the possibility of hydrogel breakdown to the tissue [34].

Although this study tried to minimize bias through a randomized clinical setting, it was impossible to construct a perfectly matched cohort of age, gender, location, and volume of edentulous area within the limited resources. Such limitation may be the cause of possible bias.

In conclusion, a self-inflating tissue expander effectively augmented soft tissue volume and both GBR and tunneling techniques confirmed their effectiveness in bone augmentation. With greater amount of bone gain and better 6 month hard tissue integrity, the TEG group (in which GBR was performed after tissue expansion) compared to the control group (with only GBR performed) showed that the combined modality of tissue expander use and GBR technique may improve the outcome and predictability of hard tissue augmentation. There was no significant difference between the TET and control group; but, considering the cost and trouble of both the operator and patient in addition to its complication rates, TET can be considered a very simple efficient and effective surgical procedure.

## Figures and Tables

**Figure 1 jcm-09-01143-f001:**
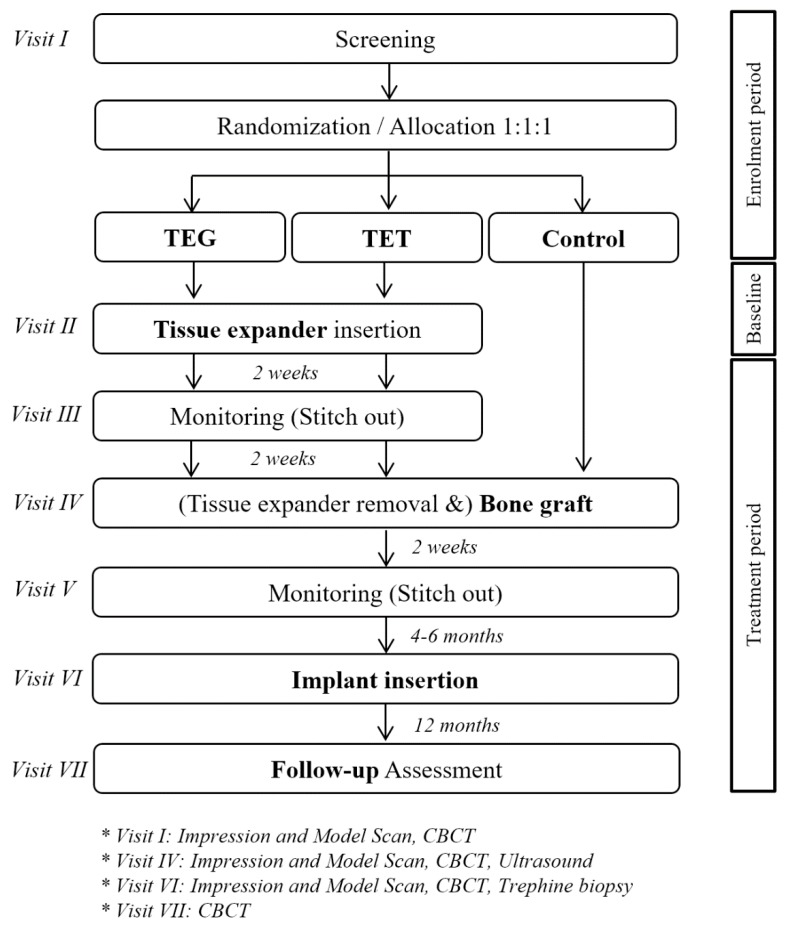
Timeline and schematic of study design and protocol. CBCT: cone beam computed tomography.

**Figure 2 jcm-09-01143-f002:**
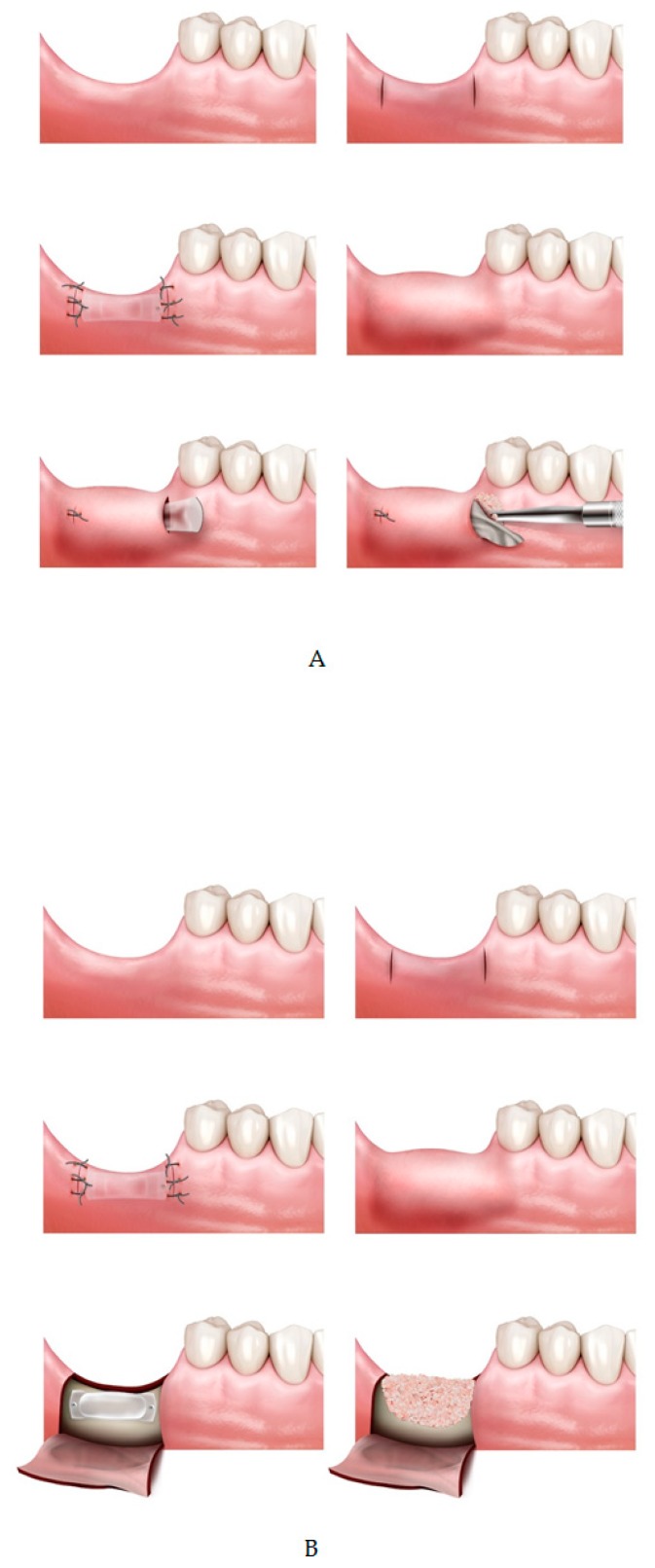
(**A**) Tissue expansion and tunneling graft (TET) group; tissue expander insertion with tunneling bone graft technique. (**B**) Tissue expansion and conventional bone graft (TEG) group; tissue expander and bone graft with conventional guided bone regeneration technique. (**C**) Control; full flap guided bone regeneration (GBR) without tissue expander insertion.

**Figure 3 jcm-09-01143-f003:**
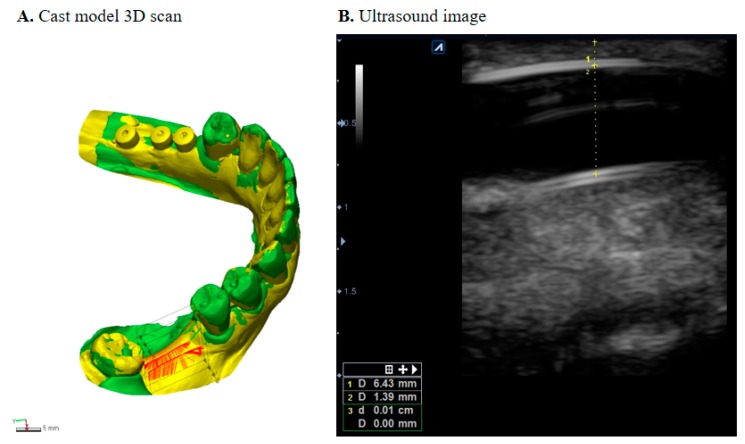
(**A**) Superimposition analysis for the measurement of soft tissue changes. (**B**) Ultrasonographic measurement of gingival thickness.

**Figure 4 jcm-09-01143-f004:**
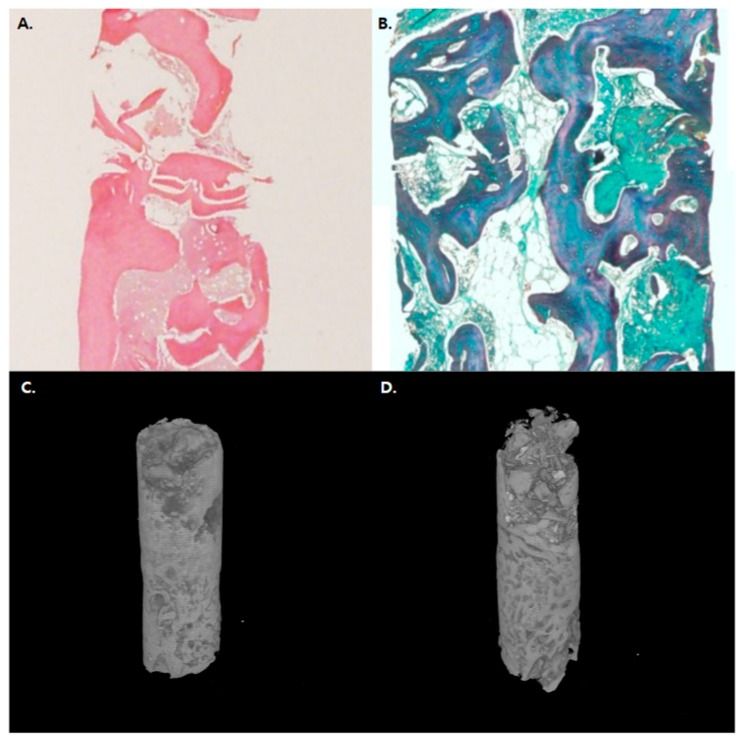
Histologic and microCT evaluation of the bone; representative histologic images of (**A**) TEG group HE staining and (**B**) control group MT staining. (**C**) 3D-reconstructed microCT image of TET group. (**D**) 3D-reconstructed microCT image of the control group showing favorable osteogenesis and osteoconduction.

**Table 1 jcm-09-01143-t001:** Descriptive statistics: age, sex, and treatment location of study participants.

Variables		Experimental		Control (*n* = 21)	Total
		TEG (*n* = 17)	TET (*n* = 19)		
Mean Age (SD)		54.35 (13.99)	57.63 (10.38)	57.86 (13.22)	56.74 (12.50)
Sex	Male	10	6	14	30
	Female	7	13	7	27
Location	Maxilla	5	13	12	30
	Mandible	12	6	9	27
Any previous surgery affecting gingival/periodontal condition^*^		6	8	7	

TEG: vertical GBR with tissue expander; TET: tunneling with tissue expander; Control: vertical GBR without tissue expander. * Includes previous bone graft history, implant surgery, any mucogingival surgery including root apicoectomy, buccal frenoplasty, periodontal flap surgery, etc. on the same site.

**Table 2 jcm-09-01143-t002:** The mean difference in soft and hard tissue measurements.

Measurments		Experimental		Control (*n* = 21)	*p* Value
		TEG (*n* = 17)	TET (*n* = 19)		
**Soft Tissue Measurements**					
Tissue Expansion	Vertical	5.42 (2.73)	5.70 (2.66)	-	n.s.
	Horizontal	6.47 (2.25)	5.49 (1.97)	-	n.s.
**Hard Tissue Measurements**					
Total amount of bone gain (Visit IV–I)	Vertical	5.71 (1.99) *	5.13 (1.32)	4.32 (0.97)	0.016
	Horizontal	5.86 (2.07)	5.03 (1.69)	4.54 (1.37)	n.s.
Bone resorption at 6 months (Visit VI–IV)	Vertical	1.16 (0.62) *	1.54 (1.12)	2.06 (1.00)	0.018
	Horizontal	0.94 (0.57) *	1.23 (0.92)	1.69 (0.81)	0.016
Peri-implant marginal resorption		0.43 (0.30)	0.36 (0.32)	0.35 (0.18)	n.s.

A t-test was used to compute the *p*-values for soft tissue measurement, and Kruskal–Wallis H test was used to compute the *p*-values for hard tissue measurements. * Indicates a significant group difference with control by post hoc test. There was no group difference between TEG and TET.

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
