# Peer review of "Tissue Expansion Improves the Outcome and Predictability for Alveolar Bone Augmentation: Prospective, Multicenter, Randomized Controlled Trial"

_jcm, 2020, doi:10.3390/jcm9041143_

Round 1
Reviewer 1 Report
“Tissue Expansion Improves the Outcome and Predictability for Alveolar Bone Augmentation: Prospective, Multicenter, Randomized Controlled Trial” presents important findings in the field of alveolar bone augmentation for preimplant surgery. This paper is well written and shows very useful and interesting results. Please refer to the following comments.
This paper shows the usefulness of Tissue expandar.
Does the preoperative evaluation of gingival condition include cases with scar formation? For example, root apicotomy, buccal frenoplasty, or flap-open extraction before tooth extraction. Please add it.
Author Response
Thank you for your important comment. The preoperative evaluation of the study included factors that could affect the integrity of the soft tissue such as the gingiva and periodontium during tissue expansion. Among them, uncontrolled severe periodontitis and gingivitis were excluded because of the study reliability and patients with the conditions you mentioned were regarded as indication for tissue expansion. As you suggested, we have included the following information as a caption under Table 1 of the Results section. Thank you.
Reviewer 2 Report
The overall review of the manuscript is positive. The work presented is well written and technically sound. Only the figures require some corrections.
The authors present 3 initial figures, namely figures “Figure 1‐2”, “Figure 1‐3” and “Figure 1‐4”. But the figure 1 caption describes also “Figure 1-1”. The authors should correct accordingly. Also the four figures should be divided. It is not correct that “Figure 1-1”, “Figure 1‐2” should belong together or even combined with the two remaining figures, “Figure 1‐3” and “Figure 1‐4”. Additionally, the figures captions should present a bit more information, namely the first 4 figures, along with some figures that are presented in the manuscript without caption (page 6)
Based on these modifications, I recommend that the paper should undergo minor revisions.
Author Response
Thank you for your comments and we apologize for our mistakes. We have clearly re-numbered all of the figures and have added some details to the caption for a clearer understanding. Previous figure 1-2, 1-3 and 1-4 were changed to Figure 2-A, B, C with revised figure captions. Please refer to the manuscript for the changes.
Reviewer 3 Report
Interesting and well written article
Author Response
Thank you so much for you kind comments and interest in our manuscript.
Reviewer 4 Report
The intended purpose of this study according to the title is to determine if tissue expansion improves vertical and horizontal bone augmentation. To accomplish this you incorporated three groups of patient. If the study was properly designed, the only variable would be the use of the expander. In your study, however, you had a number of other differences between the groups that included: different membranes, different expanders (4 types and one with 64% of the original), and dissimilar numbers of cases in the maxilla and mandible (6/13, 12/5). Other problems with the study include the following:
- You use the word "tunnel" but you made a flap and did not actually tunnel.
- With 6 different institutions participating, how did you control the similarity of the procedures and the accuracy of the measurements, s the latter where the differences were less than few millimeters?
- You did a periosteal releasing incision for the control, where no membrane was used, but not for the cases where you used a membrane. Why the difference?
- You provide actual 6 mo data on comparison of TEG and control, but not on TEG vs TET.
- You entire discussion is focused on the expanders and not on the augmentation results.
- You conclude that the results with TET and the control showed no difference, yet one should use the TET. I would seem that the results support not using the TET.
Author Response
Thank you for taking the time to review our manuscript and we greatly appreciate your insightful comments. Please find the responses to your comments numbered below.
- Thank you very much for your careful review. However, the authors are cautious about the opinion of your concerns regarding the study setting. The use of different expanders is inevitable due the differences in each patient’s size, width, and shape of the edentulous area. The Cytoplast® used in the TET group was not supported by the study protocol thus was only used under patient consent with out of pocket expenses. Unless otherwise noted, all other study patients were treated with OssMem®, which was supported by the study protocol. Both membranes are highly purified type I collagen from bovine and their difference seemed to hardly affect the results. The d-PTFE titanium reinforced membrane used in the TEG and control group is a standard membrane used in conventional guided bone regeneration techniques. Although this study tried to minimize bias through randomized clinical setting, it was impossible to construct a matched cohort of age, gender, and location within the limited resources. Possible bias regarding such limitations are discussed in the discussion section. Thank you again for your comments.
- "Although this study tried to minimize bias through randomized clinical setting, it was impossible to construct a perfectly matched cohort of age, gender, location, and volume of edentulous area within the limited resources. Such limitation may be the cause of possible bias” (Page 13, Line 352)
- The term ‘tunneling’ is not a new term created by the authors, but is a term that is used to refer to a technique that uses minimal access flap reflection compared to the square flap in conventional GBR technique. As with you, we understand that this may bring confusion to the readers, thus we cited the following article in the manuscript (Page 8, Line 151). Also, the authors revised figure 2 description to avoid readers’ confusion.
- Xuan et al. Vertical ridge augmentation using xenogenous bone blocks: a comparison between the flap and tunneling procedures. J Oral Maxillofac Surg. 2014, 72, 1660-1670.
- This was a multi-center study, but all patient results were centrally collected to one institution (Ewha Woman’s University) for analysis. The same measurements were taken by two different researchers individually and cross checked with each other for any measurement errors. Further, ICC scores were calculated at 0.81 to assure good reliability among the two scorers. The protocol for the procedure was matched and agreed in the study planning stage and the surgical procedures were filmed in the first two cases and shared during our regular meetings among the research team for further assurance.
- We apologize for the confusion. There were no differences between the TEG group and control group besides the use of expander and releasing incision, and the membrane used in the two groups were identical. We have included the following phrase for clarification.
- “Periosteal releasing incision was made to ensure primary closure of the flap in all cases of the control group. Following bone graft, a d-PTFE titanium reinforced membrane (Cytoplast® Ti-250XL; Osteogenics, NJ, USA) was placed over the grafted area before flap closure. ” (Page 8, Line 172)
- As shown in Table 2, ANOVA with post-hoc analysis between TEG, TET, and control groups was performed. Only the differences in TEG-control group were presented because the group differences between TEG-control were significant at various points but no significant group differences were found between the TEG-TET groups. To avoid any possible confusion, the Results section and Table 2 was modified as follows:
- “Control group showed higher amount of vertical (2.06 ± 1.00mm) and horizontal bone resorption (1.69 ± 0.81mm) compared to that of the TEG group (P < 0.05; 1.16 ± 0.62mm and 0.94 ± 0.57mm). TET group also showed less bone resorption, however the results were not statistically significant. (P>0.05)” (Page 11, Line 270)
- “There was no group difference between TEG and TET” (Table 2)
- As per your comments, the authors have modified and newly included the discussion on augmentation results in the 4th and 5th paragraph of the discussion section. We hope that this revision fully addresses your concern. Thank you.
- “Due to such reasons, tissue expanding GBR group (TEG) presented lower bone resorption and hard tissue retention, as shown with significant group differences in vertical and horizontal measurements 6 months post treatment.
- On the other hand, the amount of bone resorption measured 6 months after surgery was not significant in the TET and control groups. It is predicted that this is because in TEG group, ideal ossification of graft materials can be assured by the use of titanium reinforcement. However, in comparison, TET group has lower graft stability and its relative decrease in ossification is expected. In fact, in some TET group cases, a very ideal augmentation result was achieved, while in some cases, severe resorption of augmented bone was observed. Various attempts are necessary to improve the reliability of the TET technique. Considering the multiple drawbacks of the conventional GBR techniques compared to the TET method such as high failure rates, risk of dehiscence, infection and resorption, relatively high cost, long procedure time etc., the improvement of technical reliability and establishment of appropriate indication application protocol can make TET a very efficient and effective method of treatment.” (Page 13, Line 327)
6. If the same results can be obtained by the two different procedures, the comparativelysimple tunneling method, over the conventional GBR technique (control) with higher complication rates, can be used efficiently through correct indication selection. As you have commented, it is difficult to support the terms ‘more desirable’ and ‘recommended’ through this study, thus we have toned down the previous conclusion as follows:
- “There was no significant difference between the TET and control group, but considering the cost and trouble of both the operator and patient in addition to its complication rates, TET can be considered a very simple efficient and effective surgical procedure.” (Page 13, line 362)
Thank you so much for your insightful comments. I hope that we have addressed all of your concerns through this revision and we hope to hear a positive reply back from you. Thank you again.
Round 2
Reviewer 4 Report
I have reviewed your replies to my questions regarding your study. I was confused by the illustrations regarding the tunneling procedure, but it is clarified in the text so you are correct. In respect to my comment about the differences in the type of expander used, I was referring to line 142 where you state a change was made that altered the volume, and not to the various sizes and shapes that can be used. In regard to my overall criticism of the report, this has not changed. The control that was used is not appropriate for both the TET and TEG groups since the lack of an expander was not the only variable. In the control group a titanium reinforced membrane was used and this was similar to the TEG group. However, in the TET group a flexible collagen membrane was used. For a proper comparison the membrane would have to be the same in the both expansion groups.